# Research

biomechanics, physiology

stable isotopes, free flight, flight energetics, Coleoptera

**Authors for correspondence:**
Eran Levin
e-mail: levineran1@tauex.tau.ac.il
Gal Ribak
e-mail: gribak@tauex.tau.ac.il

# Insect flight metabolic rate revealed by bolus injection of the stable isotope $^{13}$C

Tomer Urca[1], Eran Levin[1,2] and Gal Ribak[1,2]

[1]School of Zoology, Faculty of Life Sciences, Tel Aviv University, Tel Aviv 6997801, Israel
[2]Steinhardt Museum of Natural History, Israel National Center for Biodiversity Studies, Tel Aviv 6997801, Israel

  TU, 0000-0001-9044-2977; EL, 0000-0002-9972-7028; GR, 0000-0002-6267-5471

Measuring metabolic rate (MR) poses a formidable challenge in free-flying insects who cannot breathe into masks or be trained to fly in controlled settings. Consequently, flight MR has been predominantly measured on hovering or tethered insects flying in closed systems. Stable isotopes such as labelled water allow measurement of MR in free-flying animals but integrates the measurement over long periods exceeding the average flight duration of insects. Here, we applied the 'bolus injection of isotopic $^{13}$C Na-bicarbonate' method to insects to measure their flight MR and report a 90% accuracy compared to respirometry. We applied the method on two beetle species, measuring MR during free flight and tethered flight in a wind tunnel. We also demonstrate the ability to repeatedly use the technique on the same individual. Therefore, the method provides a simple, reliable and accurate tool that solves a long-lasting limitation on insect flight research by enabling the measurement of MR during free flight.

## 1. Background

Insect flight is a highly demanding activity, with flight metabolic rate (MR) exceeding resting MR 100-fold in some cases [1]. As the smallest actively flying animals, insects use unique unsteady aerodynamics related to their small size and high flapping frequency [2]. They are also unique in being the only flying animals with an invertebrate body plan entailing an open circulatory system and a unique gas exchange system where oxygen is delivered directly to the flight muscles through trachea [3]. With such fundamental differences, it would be plausible to assume that insect flight physiology differs substantially from flying homeothermic vertebrate (birds and bats). However, comparing flight physiology between flying insects and vertebrates is met with the technical challenge of performing respirometry on insects during flight. Unlike flying vertebrates, where the respired air can be sampled directly via a mask placed over the mouth and nose, insects' tracheal systems limit the study of their respiration to measuring changes in $O_2$ and/or $CO_2$ levels in the surrounding medium. Such measurements are feasible in a small, closed metabolic chamber, as the insect hovers within the confined space [4,5]. However, the measurement of MR during forward flight requires the detection of small changes in gas levels from an enormous volume of air. This was performed successfully by Ellington *et al.* [6] on bumblebees in a closed wind tunnel showing that their MR hardly changed between flight speeds 0 and 4 m s$^{-1}$. Their finding suggested that theories on the relationship between power output and flight speed, developed for birds [7] inadequately explain insect flight energetics.

The use of radioactively labelled isotopes (e.g. doubly labelled water, [8]) circumvents the challenges of continuous measurement of respiration in flying animals. By injecting the flyer with the labelled metabolic substrates and measuring their body levels after activity, one can estimate their turnover and, therefore, MR. However, the measurement integrates MR over a relatively long period (hours–days), during which the animal is likely to be engaged in

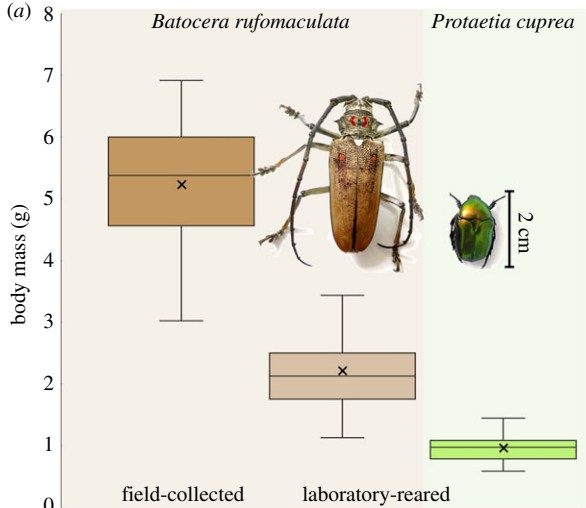

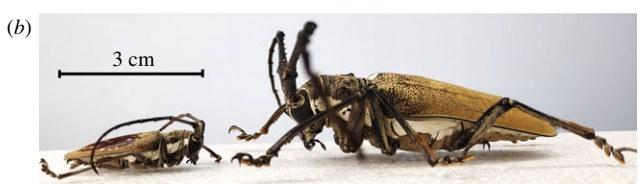

**Figure 1.** Variation in body mass of the insects used in the study. (*a*) *Batocera rufomaculata* has a high variance in adult body mass associated with differing conditions during the larval growth period. The body mass of field-collected beetles ($n = 34$, $5.23 \pm 1$ g) is significantly higher than that of laboratory-reared beetles ($n = 27$, $2.2 \pm 0.6$) (*t*-test, $p < 0.001$). The body mass of *P. cuprea* ($n = 27$, $0.97 \pm 0.22$ g) is shown in green. Boxes denote the 1st–3rd interquartile range, whiskers denote the range of observed values, the 'times' symbol denotes the mean and horizontal lines denote the median body mass. (*b*) An extreme example of intraspecific variation in adult body mass. Shown are two *B. rufomaculata* adult males. Differences in adult body mass can reach sevenfold. (Online version in colour.)

more than one activity such as rest, perching, flying and walking. By contrast, the 'bolus injection of isotopic $^{13}$C Na-bicarbonate' method [9] can measure the MR during activities lasting several minutes. The injected $^{13}$C enters the animal's bicarbonate pool and is gradually removed from the body with exhaled $CO_2$. The appearance and decline of $^{13}$C in the $CO_2$ from the metabolic chamber (measured by a carbon analyser relative to universal standard as $\delta^{13}C_{VDPB}$) is, therefore, indicative of the rate of $CO_2$ production. To date, the technique has been used to measure MR in free-flying vertebrates (birds and bats), but not on insects. Here, for the first time, we apply the method to measure the flight MR of large beetles. We used the mango stem borer (*Batocera rufomaculata*, De Geer 1775) as our model for insects relying on flight for dispersal. These large beetles are capable of prolonged flights and exhibit a high degree of intraspecific variance in adult body mass (figure 1), resulting from differences in nutrient availability during larval growth [10]. We injected beetles varying in body mass with $^{13}$C isotope and measured their MR in a metabolic chamber before and after flight. From the depletion of $\delta^{13}$C at the two measurements, we estimated the amount of $CO_2$ expired during flight. We used the technique to measure MR of the beetles during free flight and during tethered flight in a wind tunnel, simulating fast ($3.5 \text{ m s}^{-1}$) forward flight conditions. We then demonstrated the applicability of the method to other species of beetles by measuring the MR of free-flying rose chafers

(*Protaetia cuprea*) who are smaller and fly more frequently while foraging for pollen and nectar.

## 2. Methods

### (a) Animals
Mango stem borer beetles (*B. rufomaculata*) used for the establishment, calibration and validation of the technique were either taken from a laboratory population maintained at Tel Aviv University or collected in the field from infected fig trees. Laboratory-reared beetles emerged from eggs placed in individual test tubes and kept in the dark at 28°C and 70% relative humidity. The hatched larvae were fed fresh fig branches twice a week up to pupation. Experiments started at least a week after adult emergence, once the beetles began feeding. The field-collected beetles were larger than the laboratory-reared ones (mean body mass = $5.1 \pm 1.56$ and $2 \pm 0.6$ g, respectively, *t*-test, $p < 0.001$, figure 1), enabling measurement of MR as a function of intraspecific variation in body mass.

Rose chafer beetles (*P. cuprea*) were collected around Tel Aviv University during March 2021 and tested within 2 days of collection. In between, the beetles were housed in large 2.5 l transparent plastic containers with moist paper and food (apple) ad libitum.

### (b) Isotope injection
$^{13}$C-labelled Na-bicarbonate (Cambridge Isotope Laboratories Inc., Andover, MA, USA) was mixed with insect saline (mM: 150 NaCl, 5 KCl, 5 CaCl$_2$, 2 MgCl$_2$, 10 HEPES, 25 sucrose at pH 7.4) and injected through the intertergal membrane to the dorsolateral abdomen using a microsyringe (HAMILTON, Reno, NV). The molar concentration of $^{13}$C and the injected volume were determined by trial and error on 16 specimens of *B. rufomaculata* (body mass range 1.1–3.3 g) to give a peak of $\delta^{13}$C enrichment (2500–4500‰) in the metabolic chamber (chamber volume = 40 ml), within less than 10 min after the injection, and then a gradual depletion of $\delta^{13}$C in the chamber lasting for at least 30 min. The long $\delta^{13}$C depletion period was necessary to allow ample time to perform the flight experiments and return the insect to the metabolic chamber before total depletion of $^{13}$C from the beetle (figure 2). We found that injecting the beetles with a solution of 0.145 M $^{13}$C is optimal, with the injected volume (Vol) adjusted according to beetle body mass ($m$) using the equation:

$$\text{Vol } (\mu l) = 5 \cdot m(g) + 0.3. \tag{2.1}$$

### (c) Metabolic chamber and $^{13}$C measurement
Each injected beetle was placed in the metabolic chamber. Room air (25°C) was pulled by the analyser pump at a rate of 30 ml min$^{-1}$ (STP) through an Ascarite® column to scrub any atmospheric $CO_2$. The air continued throughout the chamber to a G212-*i* isotope analyser (PICARRO, Santa Clara, CA, USA). Isotope enrichment ($\delta^{13}$C) and $CO_2$ levels were measured simultaneously and automatically corrected by the analyser for H$_2$O present from the beetle's respiratory water loss. The data were recorded on a desktop computer at 1 Hz frequency.

### (d) Calibration
We calibrated $\delta^{13}$C depletion rate against $CO_2$ emission rate using 24 injected beetles (13 females varying in body mass 1.9–6.5 g; 11 males varying in body mass 1.8–6.8 g, data from all beetles were pooled together). $CO_2$ concentration and $\delta^{13}$C levels were recorded simultaneously in the metabolic chamber for approximately 2 h (the time needed for total $^{13}$C depletion from the beetles). The slope ($k_c$) of the linear decline of log ($\delta^{13}$C) with time was then correlated with the mean $CO_2$ emission rate during the same time

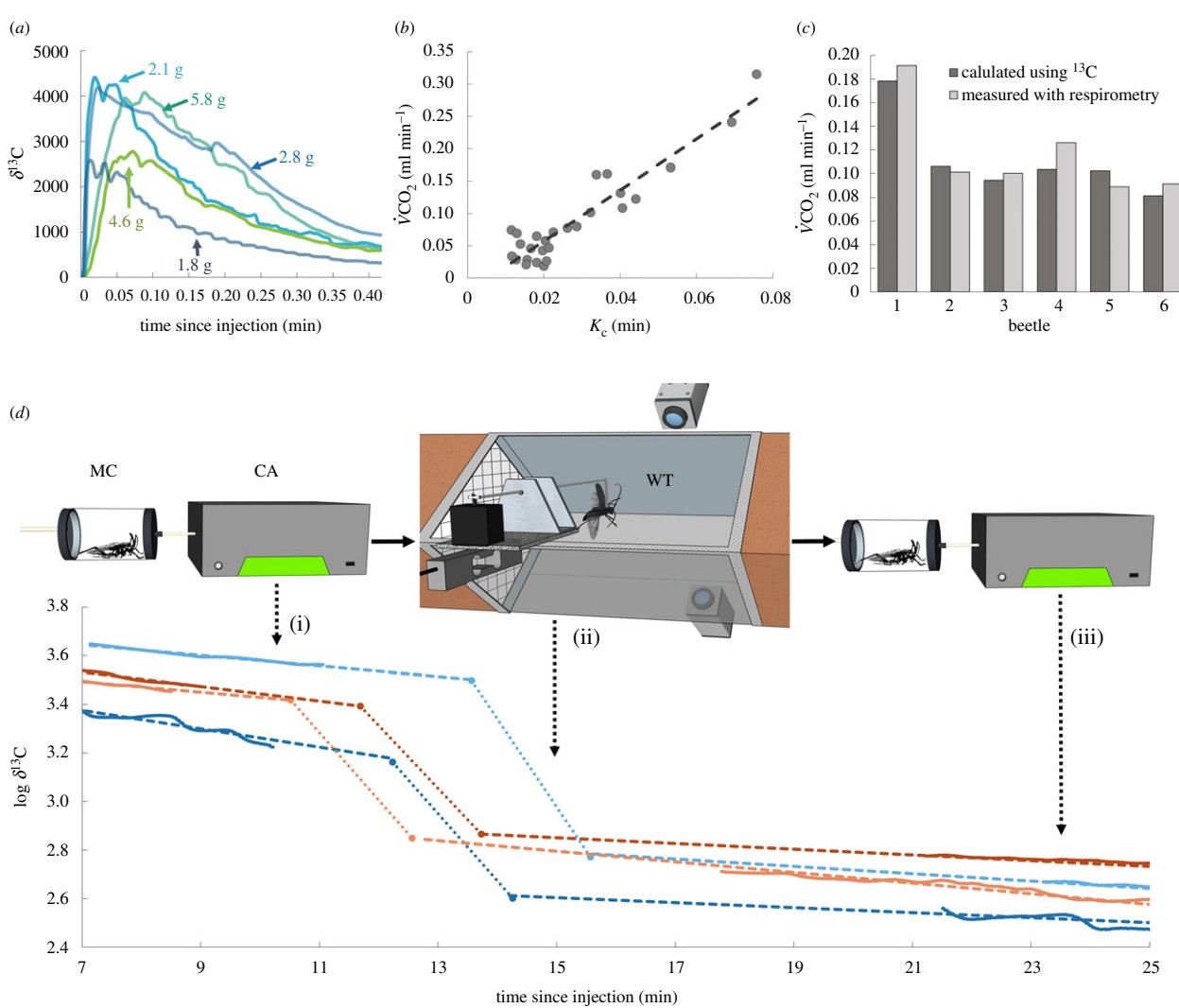

**Figure 2.** Flight MR measurement using bolus injections of $^{13}$C Na-bicarbonate. (a) Typical isotope enrichment ($\delta^{13}$C) readings of five injected beetles (*B. rufomaculta*), varying in body mass. $\delta^{13}$C first rises in the metabolic chamber, reaching a maximum within up to 10 min and then starts to decline with time. The rate of increase and decline in $\delta^{13}$C in the metabolic chamber depends on the MR of the insect and not its body mass. (b) The linear relationship between the measured logarithmic $^{13}$C-isotope elimination rate ($K_c$) and the corresponding carbon dioxide production ($\dot{V}CO_2$ ml min$^{-1}$) as the beetles are resting or harassed in the metabolic chamber ($r = 0.93$, $p < 0.001$) provide a calibration for $CO_2$ exhaled by the beetles: $\dot{V}CO_2(\text{ml min}^{-1}) = 3.9 \cdot K_c - 0.022$. The calibration was then tested on beetles harassed inside the metabolic chamber and yielded a measurement of MR with an accuracy of $89.4 \pm 5.4\%$ (c). (d) Outline of the method and calculations: $\delta^{13}$C measurements in rest before and after flight (i, iii) are carried out in a metabolic chamber (MC) connected to a carbon analyser (CA) and used to measure $^{13}$C elimination during tethered flight in a wind tunnel (ii). The decline in log-transformed $\delta^{13}$C, over time, measured before and after flight are fitted with a linear regression and is extrapolated forward and backwards to the beginning and ending of the flight bout (broken lines). The slope between the extrapolated points gives the elimination rate ($k_c$) during flight (dotted lines). 'WT', wind tunnel (flight) trials. (Online version in colour.)

period. Data from the first 5 min of the experiment were removed to allow the $CO_2$ emission measurement to stabilize.

To increase metabolic activity levels beyond resting MR, a small steel plate was glued to the prothorax of eight of the beetles, and the beetles were evoked to move by disturbing them (for 10 min) from outside the chamber using a strong magnet. We began the harassment procedure 5 min after $\delta^{13}$C began to decline in the chamber.

## (e) Validation and error estimate

Once a calibration between $k_c$ and $\dot{V}CO_2$ was established (figure 2b), we moved to validate the accuracy of the $^{13}$C technique in determining MR during activity compared to standard $CO_2$ measurements. We injected six beetles (body mass range 1.7–3.3 g) with the isotope and glued a piece of metal plate to their prothorax as described above. We allowed the beetle 10 min of rest in the metabolic chamber, during which a clear decline pattern of $\delta^{13}$C was established. The metabolic chamber

was then held vertically, evoking the beetles to climb upwards. Once reaching the top of the chamber, the beetles were dragged by a magnet to the lower end of the chamber, triggering the agitated beetles to climb up again rapidly. This 'climbing activity' was continuously repeated for 5–8 min. Subsequently, the beetles were left to rest, and the decline in $\delta^{13}$C while resting post-activity was recorded. We then estimated the MR during the activity period as in [11]: linear functions were fitted to the log-transformed measurements of $\delta^{13}$C, as a function of time during rest before and after the climbing activity. The fitted functions were then extrapolated forward and backwards to the start and end of the climbing activity bout. A third linear function was fitted between the start and end of the activity bout (e.g. figure 2d). The slope of this new function ($k_c$) was then translated into $CO_2$ emission rate using the formula obtained from the 'Calibration' section. Since the entire experiment was carried out in the metabolic chamber, we were able to compare our estimate of climbing MR using $\delta^{13}$C with the actual $\dot{V}CO_2$ measurements carried out during the climbing activity.

## (f) Tethered-flight experiments

Laboratory-reared beetles (body mass 1.1–3.3 g, $n = 16$, eight males and eight females) and field-collected beetles (body mass 3–6.9 g, $n = 17$, nine males and eight females) were injected with the isotope solution and placed in the metabolic chamber until $\log(\delta^{13}C)$ in the chamber showed a distinct trend of depletion over time (approx. 5 min after reaching maximum $\delta^{13}C$ values, figure 2d). The beetles were then taken out of the chamber and a 7 cm long (diameter = 2 mm) metal rod was glued to their dorsal prothorax using hot glue. The rod with the attached beetle was fastened vertically to a tethering system inside the working section of the wind tunnel as described by Urca et al. [12]. The tethering technique allowed the beetles to control their body angle (pitch) and the stroke plane angle relative to the oncoming wind inside the wind tunnel as their meso- and meta-thorax were free to move relative to the fixed prothorax. Preliminary free-flight experiments suggested that body angle during flight (relative to the horizontal plane) is correlated ($N = 7$, $r = 0.7$, $p < 0.01$) with flight speed, $V(\text{m s}^{-1})$ giving:

$$\text{body angle}° = -14V + 83, \tag{2.2}$$

which, for the 3.5 m s$^{-1}$ wind (flight) speed used in the wind tunnel, gives an expected body angle of 34°. The tethered beetles ($n = 33$) in this experiment assumed a mean flight body angle of $20.4° \pm 10.4°$, which was $16.2° \pm 2.6°$ lower than their body angle at rest prior to flight.

The tethered beetle started flying once the wind tunnel was turned on, and we allowed 2 min of continuous flight before the beetles were returned to the metabolic chamber and their $\delta^{13}C$ depletion and $CO_2$ emission recorded up to $^{13}C$ depletion. Flight MR was extracted as in Hambly & Voigt [11]. The decline in $\delta^{13}C$ before and after flight was used to linearly extrapolate the decline in $\delta^{13}C$ over the period of flight as described in the 'Validation and error estimate' section, but replacing the climbing trials in the metabolic chamber with tethered flight in the wind tunnel (figure 2d).

## (g) Calculation of power output during tethered flight

To estimate the power output of the beetles flying in the wind tunnel, we used two synchronized high-speed cameras (FAS-TCAM SA3_120 K, Photron, Japan) filming the beetles at 2000 fps from two different viewpoints. From the two movies, we reconstructed flapping kinematics in 3D and estimated power from the flapping kinematics using a quasi-steady blade-element model [12] (see also electronic supplementary material, S1). The estimate of mechanical power needed to move the wings through air ($P_{\text{mech}}$) enabled us to calculate the aerobic efficiency of converting metabolic work to mechanical energy. The MR measured with the $^{13}C$ technique as ml $CO_2$ s$^{-1}$ was first converted to Watts using the equation

$$P_{\text{metabolic}}(\text{W}) = \frac{\dot{V}CO_2 (\text{ml s}^{-1})}{RQ} \times (16 + 5.164 \times RQ), \tag{2.3}$$

where the second parenthesis on the right side of the equation is the oxyjoule equivalent (joules $O_2$ ml$^{-1}$) and RQ is the respiration quotient (ratio between $CO_2$ emission rate and $O_2$ intake rate, $RQ = \dot{V}CO_2/\dot{V}O_2$) [13]. RQ can range from 0.7 to 1 depending on the metabolic fuel used at the time of MR measurement. Batocera rufomaculata has a mean RQ of $1.05 \pm 0.17$ during rest ($n = 30$, T.U., unpublished data 2018) implying that carbohydrates are the main source of fuel prior to flight. For the sake of simplicity, we assumed here that carbohydrates remain the main source of fuel during short flights [14] and, therefore, used RQ = 1 in equation (2.3).

The aerobic efficiency ($\eta_{\text{aerobic}}$) can then be calculated as

$$\eta_{\text{aerobic}} = \frac{P_{\text{mech}}}{P_{\text{metabolic}}}. \tag{2.4}$$

We also assumed RQ = 1 to compare our measurements with available data from the literature (electronic supplementary material, S2).

Simultaneously with high-speed filming, the forces generated by the flying beetle were directly measured at 100 Hz using two force transducers attached to the tether arm, as described by Urca et al. [12]. The force transducers measured forces in two axes: parallel to the horizontal flight direction and vertical. The vertical force was measured directly as a decrease in the weight of the beetle during flight. The horizontal force was measured as the difference between horizontal force reading when the beetle was (i) flying and (ii) resting, with the wind tunnel operating at 3.5 m s$^{-1}$ wind speed in both cases. We used the measured forces to evaluate the tethering effect on the beetle flight. The resultant of horizontal and vertical forces ($F_{\text{total}}$, in Newton) divided by the beetle's body weight (mg) gave a non-dimensional number

$$\text{Inv} = \frac{F_{\text{total}}}{mg}, \tag{2.5}$$

where $g$ is the gravitational acceleration (9.8 m s$^{-2}$) and $m$ is beetle mass in kg. Inv scores the beetle's investment in flight: a value of Inv = 1 implies sufficient force generated to support body mass in air, whereas lower values imply insufficient force production for free-flight conditions (i.e. tethering effect).

## (h) Free-flight experiments

Three injected B. rufomaculata were treated as described above, but rather than fixed to a tether in the wind tunnel, they were placed on a wooden perch at the lobby of the Life Sciences building ($12 \times 6 \times 3$ m $L \times W \times H$). They were encouraged to take off by directing warm air from a hairdryer towards them. A stopwatch was activated each time the beetles flew, and we stimulated the beetles to fly repeatedly. The total duration of flight attempts did not exceed 8 min before returning the beetles to the metabolic chamber. The cumulative duration of each beetle's flight was used in the calculation of flight MR as described above for the wind tunnel study. Following the post-flight measurement, we allowed the three injected beetles at least an hour of rest (a duration sufficient for complete depletion of the remaining $^{13}C$ levels from the animals, as confirmed in the metabolic chamber). The beetles were then injected again and the entire flight MR measurement procedure was repeated, giving measurements of two flight bouts from each of the three beetles.

To test the technique's applicability to other species, we performed additional MR measurements on seven free-flying P. cuprea beetles. The beetles were injected with the same mass-specific dose of $^{13}C$ (equation (2.1)) and stimulated to fly freely in a large well-lit room ($7.2 \times 3.6 \times 2.5$, $L \times W \times H$ m) as described above. The calibration between $\dot{V}CO_2$ and $K_c$ for P. cuprea was established on 20 injected beetles (body mass range 0.68–1.4 g), as described in the 'Calibration' section. However, because the size of the prothorax in these beetles was much smaller compared to that of B. rufomaculata, we did not glue a metal plate on them. Rather, we increased their activity level by raising the ambient temperature (25–40°C) to give a broader range of MR for the calibration [15].

## (i) Statistics

We used the general linear model (GLM) with body mass as a covariate to compare mass-specific flight MR. Data were linearized by log transformation, and tethered flight power was corrected for tethering effects (Inv) either by using Inv as a covariate or by dividing the measured power by Inv (see specific tests below). All results are reported as mean ± s.d.

# 3. Results

## (a) Technique calibration and validation

$\dot{V}CO_2$ measured directly from injected *B. rufomaculata* in a metabolic chamber was tightly correlated with the logarithmic depletion ($k_c$) of $\delta^{13}C$ in the chamber over time ($r = 0.93$, $p < 0.001$, figure 2b). When beetles were harassed, forcing them to climb vertically inside the metabolic chamber, the $^{13}C$ Na-bicarbonate method was able to predict their climbing $\dot{V}CO_2$ with an error of $\pm 10.2 \pm 5.4\%$ compared to $\dot{V}CO_2$ measured directly ($n = 6$, figure 2c).

## (b) Tethered flight

The tethered flight MR, averaged from all the beetles, was $45.6 \pm 24.8$ ml $CO_2$ h$^{-1}$ (figure 3a). This value was, on average, $16 \pm 17.0$ fold higher than the measured resting MR of the same beetle. The tethered beetles exerted lower forces than required for supporting their body weight in the air (Inv = $0.61 \pm 0.22$). When divided by Inv to correct for tethering effects, the flight MR Inv$^{-1}$ increased to $33 \pm 47.5$ fold the resting MR.

Flight MR Inv$^{-1}$ increased with body mass (figure 3a) but did not differ between males and females ($p = 0.75$) or between laboratory-reared and field-collected beetles ($p = 0.71$) once corrected for body mass (GLM with body mass as a covariate). The data of all beetles suggested marginal hypoallometry of FMR Inv$^{-1}$ (figure 3a). The exponent of the allometric equation was 0.57 but it was not significantly different than 1.0 (confidence interval 0.13–1.0).

The mass-specific power needed to move the wings through the air ($P^*_{mech}$), estimated from the flapping kinematics (electronic supplementary material, S1), was positively correlated with Inv ($r = 0.64$, $p < 0.001$, figure 3b) but not with the mass-specific flight MR ($r = 0.12$, $p = 0.54$). $P^*_{mech}$ did not vary between laboratory-reared and field-collected beetle or between males and females (GLM with Inv as a covariate, beetle source: $p = 0.074$; Sex: $p = 0.37$, figure 3b) but did have a significant interaction between beetle source and sex ($p = 0.018$). The significance was due to field-collected males having a higher mass-specific mechanical power output than field-collected females (Tukey's *post hoc*, $p = 0.008$).

The mechanical power divided by metabolic power ($P_{mech}/MR$) gave an estimate of the aerobic efficiency ($\eta_{aerobic}$) of the tethered beetles ranging between 9 and 55%. $\eta_{aerobic}$ increased with beetle body mass ($r = 0.54$, $p = 0.003$, figure 3c).

## (c) Free-flight experiments

The three *B. rufomaculata* beetles that were injected twice and allowed to fly freely within a large room had a cumulative flight duration of 26–43 s in each of two flight bouts (electronic supplementary material, S2). The beetles survived both injections and remained vital for at least a week after the experiments. Free-flight MR exceeded the MR during tethered flight approximately threefold even after correcting for tethering effects (i.e. correcting for Inv, figure 4) and reached a mean mass-specific MR of $93.6 \pm 19.8$ ml $CO_2$ g$^{-1}$ h$^{-1}$. The smaller species of beetles (*P. cuprea*, $n = 7$) flew for 43–68 s in a flight bout and had a mean mass-specific flight MR of $69.5 \pm 30.9$ ml $CO_2$ g$^{-1}$ h$^{-1}$ (figure 4; electronic supplementary material, figure S2).

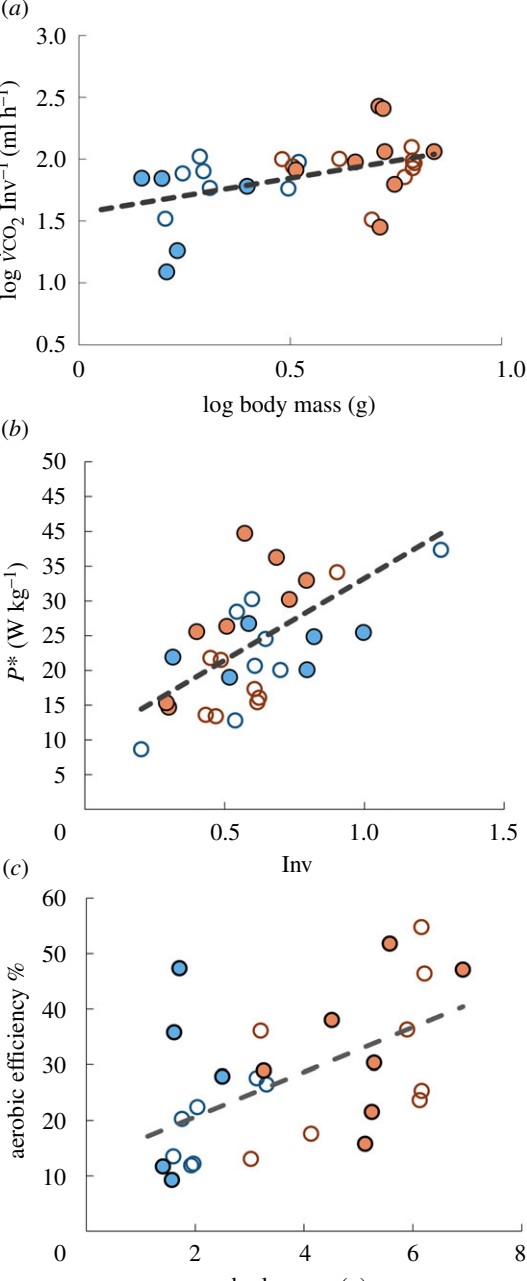

**Figure 3.** Flight energetics of *B. rufomaculata*. Males are denoted by full circles and females by empty circles. Blue and red denote laboratory-reared and field-collected beetles, respectively. (a) The log converted flight MR corrected for investment show a significant correlation with log (body mass) ($r = 0.46$, $p = 0.012$) with an allometric equation of $36 \cdot M^{0.56}$. (b) Mass-specific mechanical power increased significantly with investment ($r = 0.6$, $p < 0.001$) and showed a significant interaction between beetle source and sex (GLM, $p = 0.018$) resulting from field-collected males (full red circles) having a higher mass-specific power output compared to field-collected females (empty red circles). (c) The aerobic efficiency (mechanical power output/metabolic power) increased with body mass ($r = 0.52$; $p = 0.006$) with no significant beetle source (laboratory-reared/field-collected) or sex effects (GLM with body mass as a covariate, source: $p = 0.1$; sex: $p = 0.25$). (Online version in colour.)

# 4. Discussion

To the best of our knowledge, this is the first study showing the applicability of the $^{13}C$-bicarbonate bolus injection method for the measurement of insect flight MR. Using the method, we were able to measure the MR of *B. rufomaculata*

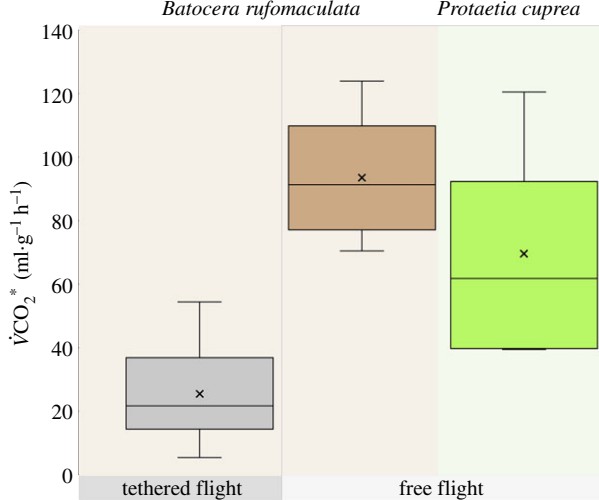

*Batocera rufomaculata*          *Protaetia cuprea*

**Figure 4.** Mass-specific MR in tethered and free flight. The mass-specific flight MR of tethered *B. rufomaculata* corrected for investment (grey, $N = 29$) was on average 3.6-fold lower than flight MR during free flight (brown, $N = 6$). The MR of free-flying *P. cuprea* (green, $N = 7$) was on average 1.34-fold lower than that of the free-flying *B. rufomaculata*. Boxes denote the 1st–3rd interquartile range, whiskers denote the range of observed values, the 'times' symbol denotes the mean and horizontal line denotes the median body mass. (Online version in colour.)

during tethered flight in a wind tunnel under conditions simulating forward flight and during free flight within a large room. Such flight conditions are not amenable to MR measurement using respirometry due to the limitation of detecting small changes in $O_2$ or $CO_2$ concentrations in the large air volumes flowing past the beetle.

The relatively large body size of *B. rufomaculata* makes it easier to inject it with the isotope solution, and we found no apparent adverse short- or long-term side effects for the animals. The injected beetles flew readily in the experiments and remained viable for four to six weeks after the injection. Moreover, we were able to inject the same beetles more than once, thus allowing us to measure their flight MR repeatedly. Hence, the technique enables repeated measurements of flight MR, enabling a comparative paired design of 'before' and 'after' treatment. Our ability to apply the method to the smaller *P. cuprea* (body mass 0.59–1.45 g) as well, suggests that the method can be applied to even smaller insects.

The $\delta^{13}C$ enrichment and depletion in the metabolic chamber as a function of time and the linear relationship between $k_c$ and $\dot{V}CO_2$ (figure 2) resembled those reported previously for vertebrates [11]. As opposed to the study of homeothermic vertebrates, where the ambient temperature was used to elevate the resting MR during calibration, we either used magnets to mechanically agitate the beetles and boost their activity levels (*B. rufomaculata*) or changing temperature (*P. cuprea*). In both cases, increasing the MR during calibration significantly improved the correlation between $k_c$ and $\dot{V}CO_2$. The inclusion of elevated activity in the calibration may be particularly important in insects where resting MR can be extremely low and highly variable. Using such calibration, the yielded accuracy of the technique for measuring MR was $89.8 \pm 5.4\%$, a relatively high accuracy considering that MR is estimated through indirect measurements of $CO_2$ production based on $^{13}C$ depletion before and after activity.

Certain specific experiments require insects to be tethered in a wind tunnel for force or electrophysiological

measurements. We were able to measure MR in such a set-up despite the high wind speed ($3.5 \text{ m s}^{-1}$) needed to simulate forward flight conditions. While tethered-flight experiments have obvious drawbacks, as revealed by the Inv < 1 found in our study, tethering the beetles allowed us to ensure constant flight conditions during continuous flight for 2 min while measuring the forces applied by the insect on the tether. Moreover, it allowed us to measure the flapping kinematics at high precision to estimate the beetle's mechanical power output during flight. Although the conditions are not representative of free flight, conducting the experiments only on free-flying beetles would have resulted in MR measurements of insects flying at different speeds, accelerating, manoeuvring and landing, thus, limiting our ability to relate the MR measurement to specific flight conditions and, therefore, making it difficult to compare the MR measurement between beetles varying in body mass. Our tethered flight set-up revealed how: (i) flight MR Inv$^{-1}$ increased with body size with no difference in mass-specific cost of flight between small and larger individuals or males and females; (ii) it also showed that larger (field-collected) males required higher mass-specific mechanical power than field-collected females (after correcting for investment) to fly in the wind tunnel at the same flight (wind) speed. Their increase in $P^*_{\text{mech}}$ was not mirrored by an increase in mass-specific flight MR contributing to a higher estimate of aerobic efficiency in larger beetles. The aerobic efficiency found for *B. rufomaculata* may be an overestimate because the metabolic power was converted to watts assuming carbohydrates were used as fuel (RQ = 1). If beetles use other substrates during flight, RQ can become as small as 0.7 [16], implying that our estimate of metabolic power may be underestimated by up to 24.5% (equation (2.3)). Higher metabolic power for the same mechanical power implies lower aerobic efficiency (equation (2.4)). More importantly, for RQ = 0.7, our flight MR data converted to $VO_2$ in electronic supplementary material, table S3 would have been approximately 25% higher. Since shifts in metabolic substrates may occur during long flight, future studies on insect flight MR should focus on determining these substrates and their RQ to allow more accurate, species-specific, assessments of MR and power.

Our free and tethered flight measurements of MR were in general agreement with previously reported values in tethered and free-flying insects (electronic supplementary material, S3). Previous studies have estimated that the lack of need to support body weight in the air during tethered flight underestimates flight MR by 20–50% [17], which agrees with our finding of a 3.6-fold higher mass-specific MR during free flight even after correcting for Inv in the tethered-flight experiment. The mean mass-specific MR measurements during free flight of *B. rufomaculata* were 35% higher than those of the rose chafer, possibly reflecting taxonomic and anatomical differences (rose chafers are more streamlined, fly with their elytra closed and have approximately threefold higher flapping frequency). Both species have mass-specific flight MR that falls within the range of reported values from other insects (electronic supplementary material, S3).

## 5. Conclusion

The development of real-time portable stable carbon isotopes analysers and the variety of labelled nutrients and molecules

has many promising directions in the study of metabolism, substrate turnover and ecophysiology of animals [18–20]. Our study demonstrates that the bolus $^{13}$C-bicarbonate injection technique can be used efficiently and effectively on flying insects under free-flight and controlled flight conditions. It thereby enables flight MR to be measured outside the confines of a closed system. The method provides a novel transformative tool for studying flight MR in insects under natural flight modes such as foraging, commuting and aerial hawking.

Data accessibility. The data are provided in the electronic supplementary material associated with this paper [21].

Authors' contributions. T.U.: conceptualization, data curation, formal analysis, investigation, methodology, software, visualization, writing—original draft, writing—review and editing; E.L.: conceptualization, data curation, methodology, project administration, resources, supervision, validation, writing—original draft, writing—review and editing; G.R.: conceptualization, data curation, methodology, project administration, resources, software, supervision, validation, visualization, writing—original draft, writing—review and editing.

All authors gave final approval for publication and agreed to be held accountable for the work performed therein.

Competing interests. We declare we have no competing interests.

Funding. We received no funding for this study.

Acknowledgements. We thank the staff of the School of Zoology at Tel Aviv University for logistical support as well as the reviewers of this manuscript for their valuable input.

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
