## [Peer Review File · Proceedings of the Royal Society B: Biological Sciences]

Review History

RSPB-2021-1082.R0 (Original submission)

Review form: Reviewer 1 (Philip Matthews)

Recommendation

Accept with minor revision (please list in comments)

Scientific importance: Is the manuscript an original and important contribution to its field?

Excellent

General interest: Is the paper of sufficient general interest?

Excellent

Quality of the paper: Is the overall quality of the paper suitable?

Excellent

Is the length of the paper justified?

Yes

Should the paper be seen by a specialist statistical reviewer?

No

Do you have any concerns about statistical analyses in this paper? If so, please specify them explicitly in your report.

No

It is a condition of publication that authors make their supporting data, code and materials available - either as supplementary material or hosted in an external repository. Please rate, if applicable, the supporting data on the following criteria.

Is it accessible?

Yes

Is it clear?

Yes

Is it adequate?

Yes

Do you have any ethical concerns with this paper?

No

Comments to the Author

This is a very nice piece of work! The basic c^{13} bolus technique appears to work very well in these beetles, and with cavity-ringdown gas analyzers becoming more commonplace, this is certainly an approach that could, at long last, move people away from relying on tethered flight mills to quantify flight performance and start asking questions about free-flying energetics. While I think that the experiments and data are generally sound, and the validations and tests are appropriate, one important issue I can see would be the reliability of the calibration curve with a changing RQ. If the insect used the same metabolic substrate (and RQ) throughout calibration and activity, then this is a non-issue. However, shifts in metabolic substrate are well known from studies on flying insects. For example, the Weis-Fogh (1952) paper you cited on locusts also indicates that during the first 30 to 90 min of flight, the RQ begins at 0.82 and falls to 0.75, while other studies have shown that glycogen, fats, and even proline are preferentially used during different phases of flight activity. Thus, the RQ of 1 that you assumed for your insects - which greatly simplifies all your calculations and assumptions - might not be the best choice when studies on other flying beetles point to the RQ being closer to 0.8 to 0.9: see Auerswald L. et al. (1998) or Thompson S.N et al. (1971). Ideally, you should measure both $V(\dot{CO}_2)$ and $V(\dot{O}_2)$ to determine RQ for your study species, and with insects as large as you were using, this wouldn't be too challenging using a closed respirometer flight mill. While this concern doesn't detract from the overall soundness of your approach for measuring free-flying $V(\dot{CO}_2)$, it is potentially a serious confounding factor when it comes to converting this respirometry data into a rate of energy use, and this should be acknowledged. The error in the final calculated power of the flying beetles could be as large as a ~24.5% underestimate if the RQ was actually 0.7 and you assumed it was 1.0, since you would have underestimated $V(\dot{O}_2)$ by 30% by assuming that $V(\dot{CO}_2) = V(\dot{O}_2)$, and then applied the oxyjoule calculation to this underestimated $V(\dot{O}_2)$, again assuming RQ = 1, and therefore the oxyjoule equivalent = 21.16 J ml⁻¹ O₂, instead of 19.61 J ml⁻¹ O₂. For this reason, calculating just how much your power estimate changes when you apply a range of RQ values to your respirometry data would be informative.

L32 No capital needed for "insects"

L54 lowercase "rufomaculata"

L95: Is the 30 ml min⁻¹ flow rate through the Picarro analyzer corrected to standard temp and pressure (STP)? I also notice that you didn't dry the excurrent air before passing it through the analyzer, presumably as any chemical drying agent would have interfered with measurement (Nafion drying columns, however, would allow respiratory water to be eliminated without

adding substantial lag to the flow). While respiratory water loss (RWL) from the insect is usually to be fairly small, there is the potential that its presence in the excurrent air would have a diluting effect on CO₂ concentration. Was RWL also considered/quantified and accounted for?

L165 RQ = “Respiratory Quotient” not “Respiration Quota”

L265 “insects”

Review form: Reviewer 2 (Marshall McCue)

Recommendation

Accept with minor revision (please list in comments)

Scientific importance: Is the manuscript an original and important contribution to its field?

Excellent

General interest: Is the paper of sufficient general interest?

Good

Quality of the paper: Is the overall quality of the paper suitable?

Excellent

Is the length of the paper justified?

Yes

Should the paper be seen by a specialist statistical reviewer?

No

Do you have any concerns about statistical analyses in this paper? If so, please specify them explicitly in your report.

No

It is a condition of publication that authors make their supporting data, code and materials available - either as supplementary material or hosted in an external repository. Please rate, if applicable, the supporting data on the following criteria.

Is it accessible?

N/A

Is it clear?

Yes

Is it adequate?

Yes

Do you have any ethical concerns with this paper?

No

Comments to the Author

This manuscript describes a series of experiments aimed at exploring and validating a new approach for measuring metabolic rates in free-flying insects. This technique has been validated for flying birds and bats over the past decades, but has never been conducted in insects – which constitute the vast majority of flying animals. This approach for small animals is only made

possible by the technological advances in real-time, laser-based isotope analyses techniques over the past few years.

In this study, the researchers used bolus injections of stable isotope (^{13}C) labeled sodium bicarbonate and tracked the rates at which this tracer leaves the body in the 'breath' as $^{13}\text{CO}_2$. They used a larger beetle species to establish tracer injection and breath measurement protocols and then further used a smaller beetle species to demonstrate/confirm proof of concept in other [smaller] insect models.

They 'calibrated' the tracer elimination models of metabolic rate by subjecting insects to different levels of non-flight activity while simultaneously measuring metabolism via indirect calorimetry (an approach that is precluded by logistics during free flying in insects). The authors concede that 'tethered' insects can be used for indirect calorimetry, and they conduct their own experiments here, but they do a good job of explaining why this is not an optimal approach for most insects.

The researchers further convert metabolic rates into units of power (i.e., Watts) which allowed them to generate estimates of aerobic efficiency that appear to be very reasonable. The authors used a level of isotope enrichment which is higher than previous experiments in birds and bats, but this works in their favor giving them a 'stronger' signal to model the decay of the ^{13}C in the breath.

The paper is generally very well written and easy to follow. The citations are a bit on the thin side, but they do include most of the seminal papers in the respective areas. The Introduction does a fine job of justifying the rationale/need for this approach and the limitations in current methodologies. The Methods section describes the experiment in sufficient detail that will allow other researchers to duplicate these experiments and apply them to new insect models. The sample sizes and statistical analyses appear appropriate. The use of allometry is refreshing and appropriate. The Results section clearly states experimental outcomes and the Supplemental documents bolster the confidence in these findings. The Discussion section is concise and emphasizes the importance of these experimental findings.

I do not have any major comments/issues to share (which is fairly uncommon in my reviews for over $n=60$ different scientific journals). I do have the following minor points that would be helpful for the authors to consider:

Explain why thermal changes can alter metabolic rates and why this was not also used as part of the isotope-elimination vs. metabolic rate calibration procedures in this study.

How soon after an initial experiment do the authors think it would be prudent to wait to conduct a second experiment?

The Final paragraph of the Discussion includes the word 'folds' which would be better replaced with the word 'fold'. This is also true for the figures and supplementary section text.

Consider referencing the two recent review papers that underscore how this approach can be more broadly incorporated into studies of physiology and ecology:

Welch Jr, Kenneth C., et al. "Carbon stable-isotope tracking in breath for comparative studies of fuel use." *Ann. NY Acad. Sci* 1365 (2016): 15-32.

McCue, Marshall D., et al. "Using stable isotope analysis to answer fundamental questions in invasion ecology: Progress and prospects." *Methods in Ecology and Evolution* 11.2 (2020): 196-214.

Figure 3 does not appear in color (in my version) and would need to include color in the final version.

Decision letter (RSPB-2021-1082.R0)

01-Jun-2021

Dear Mr Urca

I am pleased to inform you that your manuscript RSPB-2021-1082 entitled "Insect flight metabolic rate revealed by bolus injection of the stable isotope ^{13}C " has been accepted for publication in Proceedings B. Congratulations!!

The referee(s) have recommended publication, but also suggest some minor revisions to your manuscript. Therefore, I invite you to respond to the referee(s)' comments and revise your manuscript. Because the schedule for publication is very tight, it is a condition of publication that you submit the revised version of your manuscript within 7 days. If you do not think you will be able to meet this date please let us know.

I agree with the 2 reviewers and Associate Editor that this study has high potential impact.

Online supplementary material will also carry the title and description provided during submission, so please ensure these are accurate and informative. Note that the Royal Society will

not edit or typeset supplementary material and it will be hosted as provided. Please ensure that the supplementary material includes the paper details (authors, title, journal name, article DOI). Your article DOI will be 10.1098/rspb.[paper ID in form xxxx.xxxx e.g. 10.1098/rspb.2016.0049].

It is a condition of publication that data supporting your paper are made available either in the electronic supplementary material or through an appropriate repository. Please see our Data Sharing Policies <https://royalsociety.org/journals/authors/author-guidelines/#data>.

[http://datadryad.org/submit?journalID=RSPB&manu=\(Document not available\)](http://datadryad.org/submit?journalID=RSPB&manu=(Document%20not%20available)) which will take you to your unique entry in the Dryad repository. If you have already submitted your data to dryad you can make any necessary revisions to your dataset by following the above link. Please see <https://royalsociety.org/journals/ethics-policies/data-sharing-mining/> for more details.

Sincerely,

Dr John Hutchinson

Associate Editor

Comments to Author:

Associate Editor: Doug Altshuler

This is an interesting manuscript. The authors have managed to employ a stable isotope approach to measure metabolic rate in two species of beetle. The results are compared between tethered and free flight, which provides a valuable calibration. Although there are lots of studies with tethered insects, measurements of free flight metabolic rate have been lacking due to technological limitations that this study now solves. I expect this paper will be widely read by insect physiologists and that the technique will be employed by many. The work should therefore open up new possibilities in comparative physiology. It is a valuable contribution that will be appreciated by the broad readership of Proceedings B. The two referees are supportive of the

work but have a few minor suggestions, It would be helpful to see how these concerns are addressed in a revised version.

Reviewer(s)' Comments to Author:

Referee: 1

Comments to the Author(s)

This is a very nice piece of work! The basic c^{13} bolus technique appears to work very well in these beetles, and with cavity-ringdown gas analyzers becoming more commonplace, this is certainly an approach that could, at long last, move people away from relying on tethered flight mills to quantify flight performance and start asking questions about free-flying energetics. While I think that the experiments and data are generally sound, and the validations and tests are appropriate, one important issue I can see would be the reliability of the calibration curve with a changing RQ. If the insect used the same metabolic substrate (and RQ) throughout calibration and activity, then this is a non-issue. However, shifts in metabolic substrate are well known from studies on flying insects. For example, the Weis-Fogh (1952) paper you cited on locusts also indicates that during the first 30 to 90 min of flight, the RQ begins at 0.82 and falls to 0.75, while other studies have shown that glycogen, fats, and even proline are preferentially used during different phases of flight activity. Thus, the RQ of 1 that you assumed for your insects - which greatly simplifies all your calculations and assumptions - might not be the best choice when studies on other flying beetles point to the RQ being closer to 0.8 to 0.9: see Auerswald L. et al. (1998) or Thompson S.N et al. (1971). Ideally, you should measure both $V(\dot{CO}_2)$ and $V(\dot{O}_2)$ to determine RQ for your study species, and with insects as large as you were using, this wouldn't be too challenging using a closed respirometer flight mill. While this concern doesn't detract from the overall soundness of your approach for measuring free-flying $V(\dot{CO}_2)$, it is potentially a serious confounding factor when it comes to converting this respirometry data into a rate of energy use, and this should be acknowledged. The error in the final calculated power of the flying beetles could be as large as a ~24.5% underestimate if the RQ was actually 0.7 and you assumed it was 1.0, since you would have underestimated $V(\dot{O}_2)$ by 30% by assuming that $V(\dot{CO}_2) = V(\dot{O}_2)$, and then applied the oxyjoule calculation to this underestimated $V(\dot{O}_2)$, again assuming RQ = 1, and therefore the oxyjoule equivalent = 21.16 J ml⁻¹ O₂, instead of 19.61 J ml⁻¹ O₂. For this reason, calculating just how much your power estimate changes when you apply a range of RQ values to your respirometry data would be informative.

L32 No capital needed for "insects"

L54 lowercase "rufomaculata"

L95: Is the 30 ml min⁻¹ flow rate through the Picarro analyzer corrected to standard temp and pressure (STP)? I also notice that you didn't dry the excurrent air before passing it through the analyzer, presumably as any chemical drying agent would have interfered with measurement (Nafion drying columns, however, would allow respiratory water to be eliminated without adding substantial lag to the flow). While respiratory water loss (RWL) from the insect is usually to be fairly small, there is the potential that its presence in the excurrent air would have a diluting effect on CO₂ concentration. Was RWL also considered/quantified and accounted for?

L165 RQ = "Respiratory Quotient" not "Respiration Quota"

L265 "insects"

Referee: 2

Comments to the Author(s)

This manuscript describes a series of experiments aimed at exploring and validating a new approach for measuring metabolic rates in free-flying insects. This technique has been validated for flying birds and bats over the past decades, but has never been conducted in insects – which

constitute the vast majority of flying animals. This approach for small animals is only made possible by the technological advances in real-time, laser-based isotope analyses techniques over the past few years.

In this study, the researchers used bolus injections of stable isotope (^{13}C) labeled sodium bicarbonate and tracked the rates at which this tracer leaves the body in the 'breath' as $^{13}\text{CO}_2$. They used a larger beetle species to establish tracer injection and breath measurement protocols and then further used a smaller beetle species to demonstrate/confirm proof of concept in other [smaller] insect models.

They 'calibrated' the tracer elimination models of metabolic rate by subjecting insects to different levels of non-flight activity while simultaneously measuring metabolism via indirect calorimetry (an approach that is precluded by logistics during free flying in insects). The authors concede that 'tethered' insects can be used for indirect calorimetry, and they conduct their own experiments here, but they do a good job of explaining why this is not an optimal approach for most insects.

The researchers further convert metabolic rates into units of power (i.e., Watts) which allowed them to generate estimates of aerobic efficiency that appear to be very reasonable. The authors used a level of isotope enrichment which is higher than previous experiments in birds and bats, but this works in their favor giving them a 'stronger' signal to model the decay of the ^{13}C in the breath.

The paper is generally very well written and easy to follow. The citations are a bit on the thin side, but they do include most of the seminal papers in the respective areas. The Introduction does a fine job of justifying the rationale/need for this approach and the limitations in current methodologies. The Methods section describes the experiment in sufficient detail that will allow other researchers to duplicate these experiments and apply them to new insect models. The sample sizes and statistical analyses appear appropriate. The use of allometry is refreshing and appropriate. The Results section clearly states experimental outcomes and the Supplemental documents bolster the confidence in these findings. The Discussion section is concise and emphasizes the importance of these experimental findings.

I do not have any major comments/issues to share (which is fairly uncommon in my reviews for over $n=60$ different scientific journals). I do have the following minor points that would be helpful for the authors to consider:

Explain why thermal changes can alter metabolic rates and why this was not also used as part of the isotope-elimination vs. metabolic rate calibration procedures in this study.

How soon after an initial experiment do the authors think it would be prudent to wait to conduct a second experiment?

The Final paragraph of the Discussion includes the word 'folds' which would be better replaced with the word 'fold'. This is also true for the figures and supplementary section text.

Consider referencing the two recent review papers that underscore how this approach can be more broadly incorporated into studies of physiology and ecology:
 Welch Jr, Kenneth C., et al. "Carbon stable isotope tracking in breath for comparative studies of fuel use." *Ann. NY Acad. Sci* 1365 (2016): 15-32.
 McCue, Marshall D., et al. "Using stable isotope analysis to answer fundamental questions in invasion ecology: Progress and prospects." *Methods in Ecology and Evolution* 11.2 (2020): 196-214.

Figure 3 does not appear in color (in my version) and would need to include color in the final version.

Author's Response to Decision Letter for (RSPB-2021-1082.R0)

See Appendix A.

Decision letter (RSPB-2021-1082.R1)

08-Jun-2021

Dear Mr Urca

I am pleased to inform you that your manuscript entitled "Insect flight metabolic rate revealed by bolus injection of the stable isotope ^{13}C " has been accepted for publication in Proceedings B.

Your article has been estimated as being 8 pages long. Our Production Office will be able to confirm the exact length at proof stage.

Data Accessibility section

Open Access

Paper charges

Sincerely,
Editor, Proceedings B
<mailto:proceedingsb@royalsociety.org>

Appendix A

To: Professor Doug Altshuler, Associate Editor, Proceedings B

Thank you for handling our submission. The two reviewers provided helpful comments and we revised our manuscript accordingly, as detailed below. We would like to thank you and the two reviewers for the constructive and supportive review.

Referee: 1

... one important issue I can see would be the reliability of the calibration curve with a changing RQ. If the insect used the same metabolic substrate (and RQ) throughout calibration and activity, then this is a non-issue. However, shifts in metabolic substrate are well known from studies on flying insects. For example, the Weis-Fogh (1952) paper you cited on locusts also indicates that during the first 30 to 90 min of flight, the RQ begins at 0.82 and falls to 0.75, while other studies have shown that glycogen, fats, and even proline are preferentially used during different phases of flight activity. Thus, the RQ of 1 that you assumed for your insects - which greatly simplifies all your calculations and assumptions - might not be the best choice when studies on other flying beetles point to the RQ being closer to 0.8 to 0.9: see Auerswald L. et al. (1998) or Thompson S.N et al. (1971). Ideally, you should measure both $\dot{V}(\text{CO}_2)$ and $\dot{V}(\text{O}_2)$ to determine RQ for your study species, and with insects as large as you were using, this wouldn't be too challenging using a closed respirometer flight mill. While this concern doesn't detract from the overall soundness of your approach for measuring free-flying $\dot{V}(\text{CO}_2)$, it is potentially a serious confounding factor when it comes to converting this respirometry data into a rate of energy use, and this should be acknowledged. The error in the final calculated power of the flying beetles could be as large as a ~24.5% underestimate if the RQ was actually 0.7 and you assumed it was 1.0, since you would have underestimated $\dot{V}(\text{O}_2)$ by 30% by assuming that $\dot{V}(\text{CO}_2) = \dot{V}(\text{O}_2)$, and then applied the oxyjoule calculation to this underestimated $\dot{V}(\text{O}_2)$, again assuming RQ = 1, and therefore the oxyjoule equivalent = 21.16 J ml⁻¹ O₂, instead of 19.61 J ml⁻¹ O₂. For this reason, calculating just how much your power estimate changes when you apply a range of RQ values to your respirometry data would be informative.

Response: Thank you for this comment, *B. rufomaculata* has an RQ ~1 during rest and this is why we assumed this value is valid for short flights. The effect of RQ<1 on the calculations has been addressed, including a correction of equations 3 See in the method section (lines 166-168) and discussion (Lines 298-307).

L32 No capital needed for "insects"

Response: Corrected to 'insects'.

L54 lowercase "rufomaculata"

Response: Corrected.

L95: Is the 30 ml min⁻¹ flow rate through the Picarro analyzer corrected to standard temp and pressure (STP)? I also notice that you didn't dry the excurrent air before passing it through the analyzer, presumably as any chemical drying agent would have interfered with measurement (Nafion drying columns, however, would allow respiratory water to be eliminated without adding substantial lag to the flow). While respiratory water loss (RWL) from the insect is usually to be fairly small, there is the potential that its presence in the excurrent air would have a diluting effect on CO₂ concentration. Was RWL also considered/quantified and accounted for?

Response: Measurements were indeed performed under STP assumption. The RWL was taken into account with an automatic correction performed by the analyzer. This is now mentioned in lines 94-99.

L165 RQ = "Respiratory Quotient" not "Respiration Quota"

Response: Corrected.

L265 "insects"

Response: Corrected.

Referee: 2

...I do not have any major comments/issues to share (which is fairly uncommon in my reviews for over n=60 different scientific journals). I do have the following minor points that would be helpful for the authors to consider:

Explain why thermal changes can alter metabolic rates and why this was not also used as part of the isotope-elimination vs. metabolic rate calibration procedures in this study.

Response: Thermal changes were, in fact, used in this work to calibrate the relation between K_c and $\dot{V}CO_2$ for *Protaetia cuprea*. For *B. rufomaculata* we preferred to use elevated activity (exercise) which seemed more equivalent to flight. Lines 206-209 now state the reason for using ambient temperature in *P. cuprea*. It also includes a reference to the relationship between thermal changes and metabolic rates.

How soon after an initial experiment do the authors think it would be prudent to wait to conduct a second experiment?

Response: This is now addressed in lines 197-199. "We allowed the three injected beetles at least an hour of rest, a time period by the end of which ^{13}C levels in the metabolic chamber were observed to be completely depleted from the animal".

The Final paragraph of the Discussion includes the word 'folds' which would be better replaced with the word 'fold'. This is also true for the figures and supplementary section text.

Response: corrected throughout the manuscript.

Consider referencing the two recent review papers that underscore how this approach can be more broadly incorporated into studies of physiology and ecology:

Welch Jr, Kenneth C., et al. "Carbon stable-isotope tracking in breath for comparative studies of fuel use." *Ann. NY Acad. Sci* 1365 (2016): 15-32.

McCue, Marshall D., et al. "Using stable isotope analysis to answer fundamental questions in invasion ecology: Progress and prospects." *Methods in Ecology and Evolution* 11.2 (2020): 196-214.

Response: A reference to Welch et al., 2016 was added in line 323.

Figure 3 does not appear in color (in my version) and would need to include color in the final version.

Response: The uploaded figure is in colour.